# Advances in Biotechnological Production and Metabolic Regulation of *Astragalus membranaceus*

**DOI:** 10.3390/plants12091858

**Published:** 2023-04-30

**Authors:** Baoyu Ji, Liangshuang Xuan, Yunxiang Zhang, Guoqi Zhang, Jie Meng, Wenrong Mu, Jingjing Liu, Kee-Yoeup Paek, So-Young Park, Juan Wang, Wenyuan Gao

**Affiliations:** 1School of Pharmaceutical Science and Technology, Tianjin University, Tianjin 300072, China; xlpxlp@aliyun.com (B.J.); yunxiang_zhang@tju.edu.cn (Y.Z.); 18768153215@163.com (J.M.); 2School of Pharmacy, Henan University of Chinese Medicine, Zhengzhou 450046, China; m18203675687@163.com (L.X.);; 3School of Graduate, Tianjin University of Traditional Chinese Medicine, Tianjin 301617, China; 4Department of Horticultural Science, Chungbuk National University, Cheongju 28644, Republic of Korea

**Keywords:** *Astragalus membranaceus*, biotechnology, tissue culture, metabolic regulation, synthetic biology

## Abstract

Legume medicinal plants *Astragalus membranaceus* are widely used in the world and have very important economic value, ecological value, medicinal value, and ornamental value. The bioengineering technology of medicinal plants is used in the protection of endangered species, the rapid propagation of important resources, detoxification, and the improvement of degraded germplasm. Using bioengineering technology can effectively increase the content of secondary metabolites in *A. membranaceus* and improve the probability of solving the problem of medicinal plant resource shortage. In this review, we focused on biotechnological research into *A. membranaceus*, such as the latest advances in tissue culture, including callus, adventitious roots, hairy roots, suspension cells, etc., the metabolic regulation of chemical compounds in *A. membranaceus*, and the research progress on the synthetic biology of astragalosides, including the biosynthesis pathway of astragalosides, microbial transformation of astragalosides, and metabolic engineering of astragalosides. The review also looks forward to the new development trend of medicinal plant biotechnology, hoping to provide a broader development prospect for the in-depth study of medicinal plants.

## 1. Introduction

*Astragalus membranaceus* is the dry root of *Astragalus membranaceus* (Fisch.) Bge. var. *mongholicus* (Bge.) Hsiao or *Astragalus membranaceus* (Fisch.) Bge. There are more than 200 kinds of drugs which use *A. membranaceus* as a raw material. *A. membranaceus* has high utilization value, and has been used in food, dietary tonics, medicine, cosmetics (anti-aging), health care products and so on [1]. There are various chemical compounds in *A. membranaceus*, including flavonoids, polysaccharides, saponins, amino acids, sterols, and alkaloids [2]. The main active ingredients are *Astragalus* saponin (AST) and calycasin-7-glycoside (CG). According to scientific research, *A. membranaceus* has potential clinical applications for various conditions, such as heart failure [3], anemia, pneumonia, kidney disease, cancer, diabetes [4], skin problems, and reproductive system problems [5], improving immune responses [6], osteoporosis prevention and treatment, and radiation protection [7]. *Astragalus* saponins have significant medicinal value and can be used to produce cancer vaccines [8], health products, and cosmetics [9]. More than 50 saponins have been isolated from *Astragalus membranaceus* [5]. Astragalosides mainly include astragalosides I–IV, isoastragalosides I, II, IV, acetyl astragalosides, and soybean saponins [7]. The *Chinese Pharmacopoeia* stipulates that the content of astragaloside IV in *A. membranaceus* should not be less than 0.080%, and the content of calycosin-7-glucoside should be more than 0.020%. It can be seen that the content of astragaloside IV in the *A. membranaceus* root is very low.

With the improvement in people’s living standards, the demand for *Astragalus membranaceus* and *Astragalus* saponins is increasing at home and abroad [10]. *Astragalus* saponins are mainly extracted from *A. membranaceus*. As one of the most popular herbal medicines in the world, *A. membranaceus* is mainly derived from wild and cultivated resources in China. Wild plants of *A. membranaceus* are mainly produced in Inner Mongolia, Gansu, Shanxi, and Ningxia [3]. Wild resources have been over-exploited, seriously damaging the ecological environment. Cultivated plants are mainly derived from Shanxi Hunyuan, Gansu Longxi, Inner Mongolia Niuyingzi, and Shaanxi Zizhou [11]. The contents of active substances in cultivated plants are unstable and vary with changes in environmental conditions. Furthermore, infection, disease and pesticide application have also reduced the quality of cultivated *A. membranaceus*. It takes 3~4 years to collect cultivated *A. membranaceus* [12], which has increased the scarcity of medicinal plant resources and the probability of extinction of endangered medicinal plant resources [13,14]. It is of great significance to apply and develop new biotechnology to supply and replace traditional cultivation methods.

Bioengineering technology has played an important role in solving the problem of shortages of traditional Chinese medicine resources. The use of cell engineering, genetic engineering, bioreactor engineering, and biochemical engineering can quickly produce high-quality seedlings, modify the metabolic pathways of medicinal plants, and obtain active ingredients more easily, while saving the use of wild medicinal materials, which also greatly reduces production costs [15].

### 1.1. Astragalus Tissue Culture

*Astragalus* originates from the roots of *Astragalus membranaceus* (Fisch.) Ege. var. mongholicus (Ege.) Hsiao or *Astragalus membranaceus* (Fisch.) Ege. Its main chemical components are saponins and flavonoids. In recent years, plant tissue culture technology has provided conditions for the resource development of *Astragalus membranaceus*; see Table 1.

#### 1.1.1. Establishment of Regeneration Plant System of *Astragalus*

The leaves of *Astragalus* were explants for callus induction. The induction rate of the leaves on the MS + 2.0 mg/L 6-BA + 2.0 mg/L NAA medium reached 83.3%, and the hypocotyl induction rate reached 100% on the MS + 2.0 mg/L 6-BA + 2.0mg/L NAA medium [19].

#### 1.1.2. Research Progress on the Culture of Adventitious Roots of *Astragalus*

The induction of adventitious roots from stem of *Astragalus* needs callus induction first. The optimal concentrations of KT (Kinetin) and NAA are 2 mg/L and 1 mg/L, respectively. The high concentration of NAA can promote an increase in the root number, root length and number of fibrous roots, but inhibits the root thickness. Therefore, the optimal medium for inducing adventitious roots from the callus is MS + 2 mg/L KT + 2 mg/L NAA [20]. The adventitious root culture has the highest growth rate by inoculation root length 1.5 cm, inoculation volume 30 g (fresh weight)/bioreactor, and aeration volume 0.1 vvm (air volume /culture volume/min). The total polysaccharide content of adventitious root is higher than 1 yr old roots and 3 yr old plants; the total saponins and flavonoids contents are basically the same as 3-year-old roots and higher than 1-year-old roots [12].

#### 1.1.3. Research Progress of the Hairy Roots Culture of *Astragalus*

Some researchers have screened the hairy root line AMRRL VI and AMHRL II with high saponins and isoflavone content, respectively, and optimized the culture temperature of 27.8 °C, inoculation volume of 1.54%, sucrose concentration of 3.24% and culture time of 36 days [21]. The yield of a 30 L culture of *Astragalus* hairy roots for 20 days is similar to a 250 mL and 1 L shake flask culture, but higher than the 10 L bioreactor [22]. Hairy roots can grow rapidly and have genetic and biochemical stability, high productivity, and adaptability for large-scale systems [23].

#### 1.1.4. Research Progress on the Culture of *Astragalus* Suspension Cells

The best explant for the growth of *Astragalus* suspension cells is the hypocotyl, and the optimal culture conditions are MS + 1.5 mg/L 2,4-D + 2.0 mg / L 6-BA + 3% sucrose, pH:5.5–6.0. The initial inoculation volume is 50 g/L, and the subculture inoculation volume is 37 g/L. The growth characteristics of *Astragalus* suspension cells have been studied; the growth period is 25 d. In detail, the first ten days are the lag period, and 11–19 d is the logarithmic phase. The cell biomass reached its maximum at 19 days, 20–22 days for the stationary phase, and 23–25 days for the decay period. The cell growth curve was drawn. This was the first time that the *Astragalus* cell suspension system has been established. The growth curve of the suspension cell culture was nearly an “s” shape, and its growth cycle was about 14 days. The best harvest time for the cell culture to produce *Astragalus* polysaccharides, saponins and flavonoids was the 15th day [24].

#### 1.1.5. Culture of *Astragalus* Protoplasts

The preparation of protoplasts of *Astragalus* leaves is far easier for *Astragalus* callus, and it can obtain a large number of highly viable protoplasts. In this method, the explant can be hydrolyzed by a mixed enzyme of 2% cellulase + 0.5% hemicellulose + 0.5% pectinase for 12 h. Then, the high-quality *Astragalus* protoplasts will be separated more efficiently [25].

### 1.2. Astragalus Metabolism Control Research

Plants and plant cells, tissues and organs in vitro, with physiological and morphological responses to microbial, physical, and chemical factors, are called elicitors. Arousal is the process by which plants induce or enhance the synthesis of secondary metabolites to ensure their persistence and competitiveness [26]. The application of elicitors is the focus of current research, and has been considered as one of the most effective methods to improve the synthesis of secondary metabolites in medicinal plants [27]. At present, some hypotheses about the mechanisms of the elicitor-promoting plant secondary metabolism have been raised. As an exogenous signal, an elicitor can cause the recognition of receptor sites on the plant cell membrane. The combination of the two causes physiological and biochemical chain reactions in the cell membrane and in the cell, which changes the permeability of the cell membrane to specific molecules, and changes the chemical signal molecules, enzyme activity, and gene expression, which are related to the synthesis of secondary metabolites. All of these will eventually lead to the synthesis and accumulation of active ingredients [28]. The elicitor can be a compound that stimulates any type of physiological abnormality in plants [29].

Elicitors can be divided into abiotic types, which are compounds that stimulate any type of physiological abnormality in plants [26], physical induction, such as trauma, waterlogging, high temperature, light, etc., and biological type (bacteria, fungi and their extracts, such as *Saccharomyces cerevisiae*, *Pichia pastoris*, chitosan, cell wall polysaccharides, and plant endophytes) [27].

For a study on the metabolism control of *Astragalus*, see Table 2. The effects of inducing factors on the secondary metabolites of *Astragalus* are shown in Table 3.

#### 1.2.1. *Astragalus* Saponins

In order to increase the production of astragaloside IV (AGIV) in hairy root cultures (AMHRCs) of *Astragalus*, a new method combining deacetylated biocatalysis and the induction of immobilized penicillium canescens (IPC) was proposed. The AMHRCs and IPC were co-cultured for 60 h, and the content of AGIV was 14.59 times higher than the control group. In addition, under the induction of IPC, the expression of AGIV biosynthetic pathway-related genes was significantly up-regulated. This method provides an effective and sustainable approach for the large-scale production of AGIV [39]. The accumulation of *Astragalus* metabolic components is affected by many factors. Yeast extract (10g/L) was added on the 13th day of the *Astragalus* suspension cell culture. The PAL activity was upregulated after 36h, and the total phenol content reached the maximum after 48 h. The maximum PAL activity and total phenol content were 2.88 times and 2.12 times higher than the control, respectively. Therefore, the PAL activity induced by yeast extract was positively correlated with the accumulation of total phenols. It is believed that after adding yeast extract, the early defense response of *Astragalus* cells to biological and abiotic environmental stressors is up-regulated. Therefore, the phenolic branch of secondary metabolism is up-regulated [30]. Hexanal can significantly affect the growth of the *Astragalus* adventitious root at solid culture condition, 10 μmol/L n-hexanal has an inhibitory effect, and 50 μmol/L has a promoting effect; 50 μmol/L n-hexanal in a liquid culture system can promote the synthesis of saponins in *Astragalus* adventitious roots [40].

Methyl jasmonate (MJ), acetylsalicylic acid (ASA) and salicylic acid (SA) can be used to induce and regulate the biosynthesis of saponins and isoflavones in the hairy root of *Astragalus*. Under the condition of an MJ concentration of 157.4 μM and an induction time of 18.4 h, the total saponin yield was 2.07 times that of the control. It was also revealed that *MVD*, *IDI*, *FPS*, and *SS* are the key enzyme genes in the pathway of MJ-induced regulation of astragaloside biosynthesis [21]. To study the response of the hairy root of *Astragalus* to 100 μM methyl jasmonate (MeJA), DESeq analysis showed that 2127 genes were up-regulated and 1247 genes were down-regulated, and among the 2,127 up-regulated genes 17 were new astragaloside biosynthesis genes and 7 were isolated new genes for the biosynthesis of isoflavones and isoflavone glucoside-7-O-β-D-glucoside. The accumulation of ASTS, Mucor, and CG in the hairy roots treated with MeJA increased significantly [32].

Exogenous transcription factors also have a regulatory effect on the synthesis of astragalosides. Li et al. (2022) [41] found that Arabidopsis MYB12, anthocyanin pigment 1 (PAP1), and maize leaf color (LC) transcription factors had regulatory effects on the synthesis of astragaloside metabolites. The overexpression of LC led to the accumulation of astragaloside I-IV. The accumulation of astragaloside I and IV was higher. The overexpression of MYB12 increased the accumulation of astragaloside I in transgenic hairy roots, followed by astragaloside IV. The overexpression of PAP1 increased the synthesis of astragaloside I and IV. Several key genes in the biosynthesis pathway of valeric acid, especially 3-hydroxy-3-methylglutaryl coenzyme A reductase (HMGR1, HMGR2 and HMGR3), were differentially up-regulated in response to these transcription factors, which could lead to the synthesis of astragaloside IV in the hairy roots of membranous *Astragalus membranaceus*.

Real-time fluorescence quantitative PCR analysis of eight key enzymes in the synthetic pathway of astragaloside acetyl-CoA acetyltransferase(AATC), 3-hydroxy-3-methylglutaryl-CoA synthase (HMGS), 3-hydroxy-3-Methylglutaryl coenzyme A reductase (HMGR), isoprene pyrophosphate isomerase (IDI), farnesyl pyrophosphate synthase (FPS), squalene synthase (SS), squalene epoxy Enzyme (SE), and Cycloaltinane synthase (CAS) gene expression levels, found that in HMGR, FPS, SE, and CAS genes under the water regulation of *Astragalus* saponins content, the regulation effect was obviously the main regulatory gene [31]. Real-time quantitative polymerase chain reaction (QRT-PCR) technology was used to study the expression levels of genes related to the AST biosynthetic pathway in *Astragalus* plant seedling roots (SRS), adventitious roots (ARs), and hairy roots (HRs). It was found that genes involved in the AST biosynthetic pathway had the lowest transcription levels in ARs, and showed similar patterns in HRs and SRs. ARs and CG had higher phenylalanine concentrations, and phenylalanine was the predecessor of the phenylpropane biosynthesis pathway [42]. Therefore, both biological and abiotic factors can affect the metabolic regulation of the active ingredient of *Astragalus*.

#### 1.2.2. *Astragalus* Flavone

Isoflavonoid and Calycosin-7-glucoside (CGs) are accumulated in the whole plant of *Astragalus*, and mainly concentrated in the leaves. It was found that after 10 days of the UV-B treatment of *Astragalus* root, the change in isoflavone content in the *Astragalus* root was positively correlated with the expression of isoflavone biosynthesis related genes, and UV-B radiation significantly induced the isoflavone synthesis, which also provided a feasible heuristic strategy for understanding the accumulation of isoflavones in *Astragalus* [35]. Using ultraviolet light (UV-A, UV-B and UV-C) to promote the accumulation of isoflavones in AMHRCs, it was found that 86.4 kJ/m (2) UV-B had the best effect of promoting the production of isoflavones in AMHRCs at 34d. UV-B up-regulated the transcriptional expression of all genes involved in the isoflavone biosynthesis pathway. According to the results, PAL and C4H are two potential key genes that control isoflavone biosynthesis [36]. Using MJ, ASA, and SA signaling molecules to induce and regulate the biosynthesis of isoflavone secondary metabolic active components in *Astragalus* hairy roots, it was found that the yield of total isoflavones with an MJ concentration of 283 μM and induction time of 33.75 h was 9.71 times that of the blank control group. It was also revealed that CHI and IFS are the key enzyme genes in the pathway of MJ-induced regulation of *Astragalus* isoflavone biosynthesis [21]. *Astragalus* hairy root cultures (AMHRCs) were co-cultured with immobilized food-grade fungi to increase the production of trichomes (CA) and formononetin (FO), and 34-day-old AMHRCs were immobilized with *immobilized Aspergillus niger* (IAN) for 54 h of co-cultivation. In the highest accumulation of CA and FO, IAN induction can promote the generation of endogenous signaling molecules involved in plant defense responses, resulting in the significant upregulation of CA and FO biosynthetic pathway gene expressions, thereby increasing the production of CA and FO [34].

By studying the changes in CGs content and the expression of related genes under different conditions, including phenylalanine ammonia lyase (PAL1), cinnamic acid 4-hydroxylase (C4H), chalcone synthase (CHS), chalcone reductase (CHR), chalcone isomerase (CHI), isoflavone synthase (IFS), and isoflavone 3’-hydroxylase (I3’H) expression, the effects of different conditions on CGs biosynthesis were found. These seven genes’ expression levels showed a light-dependent manner under low temperature stress, but they showed different expression patterns when *Astragalus* plants were transferred from 16 °C to 2 °C or 25 °C or 2 °C (maintained for 24 h) to 25 °C. There were different effects on the transcription of PAL1 and C4H in the phenylpropanoid pathway in leaves. The expression of PAL1 changed markedly, which was consistent with the change in CGs content. PAL enzyme activity seemed to be the limiting factor in determining CGs levels. The PAL enzyme inhibitor L-alpha-aminooxy-beta-phenylpropionic acid almost completely blocked the accumulation of CGs at low temperatures. This confirmed that PAL1, as an intelligent gene switch, directly controls the accumulation of CGs in a light-dependent manner during low temperature treatment [33]. PAL may be a key point for flux into flavonoid biosynthesis in the genetic control of secondary metabolisms in *Astragalus Mongholicus* [43]. A new AmCHR gene was cloned from *Astragalus*, which is a new member of the CHR gene family. It is speculated that the expression of AmCHR is closely related to the accumulation of isoflavone glucoside in *Astragalus* [44]. Chalcone synthase (CHS) is a key enzyme and rate-limiting enzyme for the biosynthesis of flavonoids. The AnCHS gene was cloned from *Astragalus*, and proved that the expression of AnCHS and the accumulation of *Astragalus* isoflavones are closely related [45].

#### 1.2.3. *Astragalus* Polysaccharide

*Astragalus* polysaccharide is the main component of *Astragalus*, which is composed of hexuronic acid, glucose, fructose, rhamnose, arabinose, galacturonic acid, and glucuronic acid, etc. It can be used as an immune promoter or regulator, and has anti-virus, anti-tumor, anti-aging, anti-radiation, anti-stress, and anti-oxidation effects, among others. The pathway of secondary metabolite biosynthesis can be explained by the identification of candidate genes and important regulatory factors. The best inducer for screening the *Astragalus* polysaccharide metabolic pathway is silver nitrate solution, the best treatment site is the underground part, the best treatment time is 6–9 days, and the best polysaccharide content change detection site is the *Astragalus* root; 36 unigenes were found in the metabolic pathway of the polysaccharides metabolized [46]. Studies have found that methyl jasmonate, salicylic acid, IAA, and NAA promote the accumulation of *Astragalus* hairy root polysaccharides and total saponins, and IBA and 2.4-D have a negative effect on the accumulation of *Astragalus* hairy root polysaccharides and total saponins [47]. This provides theoretical guidance and technical support for regulating secondary metabolic pathways and increasing the content of *Astragalus* secondary metabolites by genetic engineering.

The study on the regulation of secondary metabolites of *Astragalus* is shown in Figure 1.

3-hydroxy-3-methylglutaryl coenzyme A reductase (HMGR), farnesyl pyrophosphate synthase (FPS), squalene epoxy enzyme (SE), cycloaltinane synthase (CAS), methyl jasmonate(MJ), salicylic acid(SA), chalcone synthase (CHS), chalcone Reductase (CHR), chalcone isomerase (CHI), isoflavone synthase (IFS) and isoflavone 3’-hydroxylase (I3’H), phenylalanine ammonia (PAL), cinnamic acid 4-hydroxylase (C4H), farnesyl pyrophosphate synthase (FPS), squalene synthase (SS), isoprene pyrophosphate isomerase (IDI), mevalonate-5-diphosphate decarboxylase (MVD).

### 1.3. Research Progress on Synthetic Biological Pathways of Astragalosides

#### 1.3.1. Research Progress on the Biosynthesis Pathway of *Astragalosides*

Astragalosides are the main active ingredient of *Astragalus*; they belong to triterpenoid saponins and are extremely important secondary metabolites in *Astragalus.* They are similar to the biosynthesis of other triterpenoid saponins. The biosynthetic pathway of astragalosides in plants includes the mevalonate (MVA) pathway and the 2-C-methl-D-erythritol-4-phospate (MEP) pathway. These two pathways ultimately produce the precursor isopentenyl pyrophosphate (IPP). IPP is catalyzed by farnesyl pyrophosphate synthase (FPS) to produce farnesyl diphosphate (FPP), FPP is catalyzed by squalene synthase (SS) to produce squalene, squalene is catalyzed by squalene epoxidase (SE) to produce 2,3-oxidosqualene, OS), and 2,3-oxidosqualene is catalyzed by cycloartenol synthase (CAS) to produce cycloartenol, which is the precursor of triterpene saponins. The cycloastragenol biosynthesis pathway can be seen in Figure 2. The functions of cytochrome P450 (CYP450), glycosyltransferases, and other genes required for the downstream synthesis pathway of astragalosides are being analyzed [48].

Cycloartenol synthase (CAS) has a typical 9, 19—cyclopropane moiety. Astragalosides are gradually generated by cycloastragenol-type saponins in the downstream pathway. Cycloastragenol, which was further synthesized from cycloartenol, has a 20, 24-epoxy ring, and C6, C16, and C25 hydroxyl groups. Then, structurally diverse glycosides synthesize different astragalosides through a variety of glycosylation modes. This includes xylose, glucose, and glycosyl parts of single-chain, double-chain, triple-chain, or branched chains. The sites of action are 3-OH, 6-OH, 25-OH, and 20-OH. CAS converts the 2,3-oxidosqualene skeleton into a chair–boat–chair conformation, and then several specific CYP450s may catalyze the conversion of cycloartenol into cycloastragenol. CYP450 constitutes an important, highly differentiated sequence superfamily, which can be divided into 10 families. In these families, the CYP72 family is involved in the catabolism of isoprenoid hormones. The CYP71 family modified shikimic acid products and intermediates. The CYP85 family is involved in the modification of cyclic terpenes and sterols in the brassinolide pathway. Further studies have identified CYP93E1 from Leguminosae (CYP71 family) and CYP88D6 from *Glycyrrhiza* (CYP85 family), both of which are involved in the biosynthesis of saponins. Therefore, CYP71, CYP72, and CYP85 families may be involved in the biosynthesis of astragalosides [49].

Chen et al. (2023) [50] found that AmOSC3 is acycloartenol synthase expressed in both aerial and underground parts. It is related to the synthesis ofastragalosides (cycloartane-type) in the roots. Jing Chen et al. [49] found a high-quality CAS transcript by transcriptome analysis of *A.mongholicus*, and detected 22 CYP450s related to the synthesis pathway of astragalosides. Seven transcripts belong to the CYP71 family (P57867.0, P5215.0, P7366.0, P50274.0, P26377.0, P71378.0, and P60800.0). One transcript belongs to the CYP72 family (P71249.0). Nine transcripts belong to the CYP85 family (P52746.0, P50482.0, P60763.0, P51105.0, P65633.0, P32417.0, P69412.0, P69412.2, and P43338.0). In addition, 25 UGT genes were detected to be related to the downstream synthesis pathway of astragaloside. Liu Lu et al. [51], based on transcriptome data, screened fourteen CYP450 genes with up-regulated expression, of which nine genes, four genes, and one gene were relatively highly expressed in roots, leaves, and stems, respectively. Duan et al. (2023) [52] found, for the first time, that AmCAS1 (a cycloartenol synthase) can catalyze the conversion of 2,3-oxidosqualene to cycloartenol, by transcriptome and phylogenetic analysis. Four glycosyltransferases were screened out, which were AmUGT15, AmUGT14, AmUGT13, and AmUGT7. They can catalyze the biosynthesis of cycloastragenol saponins, and the catalytic functions are 3-O-xylosylation, 3-O-glucosylation, 25-O-glucosylation/25-O-xylosylation, and 20-O-glucosylation, respectively. Among them, AmUGT15 can catalyze the conversion of cycloastragenol-6-O-D-glucoside to astragaloside IV or catalyze cycloastragenol into cycloastragenol-3-O-D-xyloseside; AmUGT13 can catalyze the conversion of astragaloside IV to Astragaloside VII or cycloastragenol-3-O-D-xylside to isostragaloside IV; AmUGT7 catalyzes the conversion of cycloastragenol-3-O-D-glucoside to cycloaraloside; and AmUGT14 can catalyze the conversion of cycloastragenol to cycloastragenol-3-O-D-glucoside. The Astragaloside biosynthesis pathway can be seen in Figure 3.

#### 1.3.2. *Astragalus* Saponin Metabolism Engineering Research

A pair of special primers were designed with the cloned starch grains and starch synthase (GBSS) gene sequence from the hairy roots of *Astragalus* to construct the expression vector PbI-GBSS. It was confirmed that the enzyme activity of GBSS gene-transformed *Escherichia coli* was 20% higher than that of the untransformed strain [53]. The total RNA of *Astragalus* leaves was extracted by reverse transcription to synthesize cDNA, which was cloned into pPIC9K by in vitro homologous recombination technology to construct the pPIC9K-PAL expression vector. Then, it was transformed into *Pichia pastoris* GS115 to obtain a relatively pure phenylpropane amino acid ammonia lyase. The content of phenylalanine ammonia lyase after purification accounted for 11.54% of the total protein, and the highest specific activity reached 4270 U/mg [54]. At present, there has not been much research on the synthetic biology of *Astragalus*, which still needs to be explored continuously.

#### 1.3.3. Biotransformation of *Astragalosides*

The biotransformation methods of astragalosides include microbial transformation and enzymatic hydrolysis. In recent years, the biotransformation of cycloastragenol saponins has achieved many research results.

##### Microbial Transformation

The microbial transformation of astragalosides mainly uses enzymes in microorganisms to convert raw materials into required target products through complex and special metabolic pathways. Fungi are commonly used microorganisms for the biotransformation of astragalosides. Eight different yeast strains, such as *Aspergillus niger*, *Aspergillus oryzae* and white rot fungi, were selected for the biotransformation of astragaloside IV. It was found that *Aspergillus niger* had the strongest biotransformation activity. After transformation, the content of astragaloside IV increased by 10.7 times and reached 2.326 mg/g [55]. Li Ye [56] selected the strain *Absidia corymbifera* AS2 to transform astragalosides to ASI. This strain enhanced ASI production approximately fourfold when cultures were supplemented with 5 g/L of crude. Bedir et al., (2015) [57] studied the microbial transformation of *Astragalus* derived sapogenins, namely, *Cycloastragenol*, *astragenol*, and *Cyclocanthogenol*, by *Cunninghamella blakesleeana* NRRL 1369 and Glomerella fusarioides ATCC 9552. The unique enzyme system of both fungi resulted in hydroxylation, cyclization, dehydrogenation, and oxidation reactions. Under the deacetylation of fungal endophyte Penicillium canescens, which were isolated from *pigeon pea*, a novel and highly efficient biotransformation method of astragalosides to astragaloside IV in Radix Astragali was investigated [51]. Meng et al.(2018) [53] studied the biotransformation characteristics of astragaloside components in human intestinal flora and obtained four products and determined their structures: astragaloside I (AS-I), astragaloside II (AS-II), astragaloside III (AS-III), and astragaloside A (AS-IV). A novel biotechnology approach of combining deacetylation biocatalysis with the elicitation of IPC in *Astragalus membranaceus* hairy root cultures (AMHRCs) was proposed for the elevated production of astragaloside IV (AG IV). The highest AG IV accumulation was achieved in 36-day-old AMHRCs co-cultured with IPC for 60 h, which resulted in the enhanced production of AG IV by 14.59-fold in comparison with that in the control (0.193 ± 0.007 mg/g DW).

##### Enzymolysis

The most commonly used enzymes are β-glucosidase and β-xylosidase. Commercial β-glucosidase can hydrolyze astragaloside IV into cycloastragenol-6-O-β-D-glucoside; β-Glucosidases and β-xylosidases are two categories of enzymes that could cleave out nonreducing, terminal β-D-glucosyl and β-D-xylosyl residues with the release of D-glucose and Dxylose [58]. Qi Li et al.(2019) [58] used the purified thermostable and sugar-tolerant enzymes from *Dictyoglomus thermophilum* to hydrolyze ASI synergistically, which provided a specific, environmentally friendly and cost-effective way to produce CAG. An acetyl esterase from *A. corymbifera* AS2 was purified and its catalytic pathways were investigated, which showed unique enzymatic characteristics and enabled the clarification of the biotransformation pathways of astragalosides [59]. A novel β-glucosidase from *Phycicoccus* sp. Soil748 (Bgps) was discovered, possessing the efficient conversion rate for cycloastragenol-6-O-β-D-glucoside (CMG) into Cycloastragenol (CA). The optimum temperature and pH value of Bgps were determined as 45 °C and 7.0 [60]. It can be seen that the strains with high conversion efficiency were excavated from a variety of microorganisms, and the highly specific and efficient enzymes were isolated and purified. After large-scale expression in vitro, the development of commercial invertase is helpful for large-scale production [48].

## 2. Conclusions

As the leading industry in the world today, bioengineering technology has a high utilization rate in many industries such as medicine, agriculture, the chemical industry, and food. It has a significant role in promoting economic development, and also provides valuable technology for the development of medicine and information resources. Using bioengineering technology to produce medicinal plant active ingredients is an effective way to solve the problem of Chinese medicine resource shortages. With the development of synthetic biology, metabolic engineering, and protein engineering, the application of new bioengineering technology for medicinal plant resources’ reasonable utilization and in-depth development has great development potential. Based on the traditional literature and modern biotechnology research, this article summarizes the tissue culture and metabolic regulation of active ingredients and the synthetic biology of *Astragalus*.

However, the synthesis mechanisms of some biologically active ingredients are still unclear, and there is a lack of information on the biosynthetic pathways and the complex regulatory mechanisms of the biosynthesis of target compounds. There are still key steps in the biosynthetic pathways of cycloastragenol and astragaloside IV that have not been resolved, which also thwarts the biosynthesis process of astragaloside. It can be seen that the metabolic pathways of secondary metabolites are still in need of a breakthrough in biotechnology research. Therefore, it is urgent to conduct more in-depth studies on these active ingredients and study new biotechnology. Clarifying medicinal value and supporting the further development of new products will be the focus of future research. This paper is helpful for the further study of the production and biotechnology of effective components in *A. membranaceus*.

## Figures and Tables

**Figure 1 plants-12-01858-f001:**
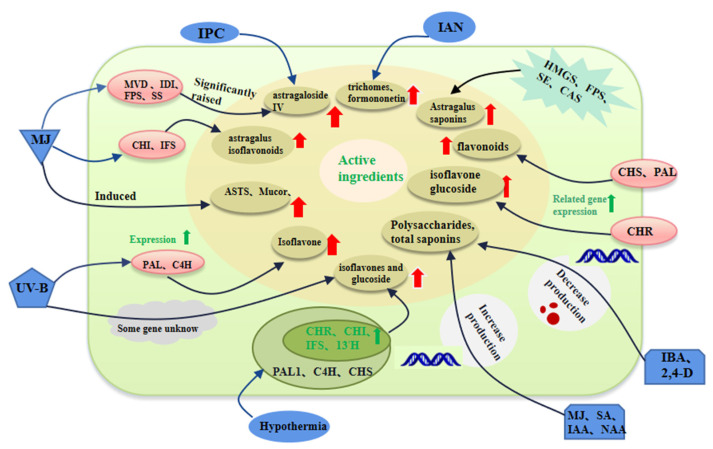
Study on the regulation of secondary metabolites of *Astragalus.* Note: The blue boxes represent biological and abiotic elicitors, the pink boxes represent genes, and the brown boxes represent secondary metabolite types.

**Figure 2 plants-12-01858-f002:**
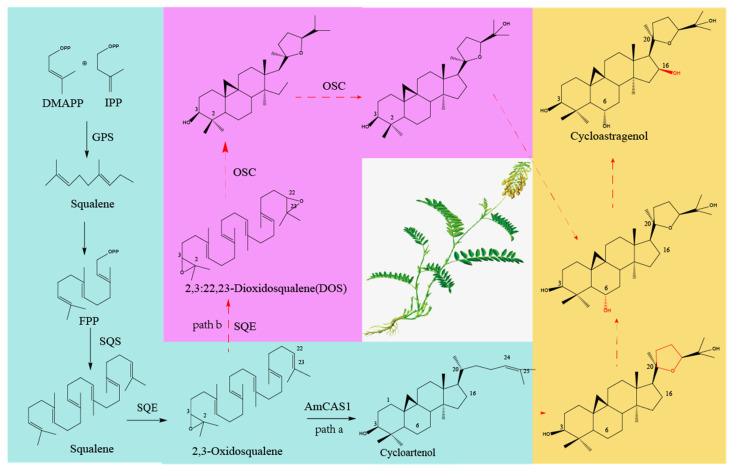
Cycloastragenol biosynthesis pathway. Note: The blue part represents the resolved pathway, and the orange and purple parts represent the speculative pathway.The red dotted line represents the speculative path.

**Figure 3 plants-12-01858-f003:**
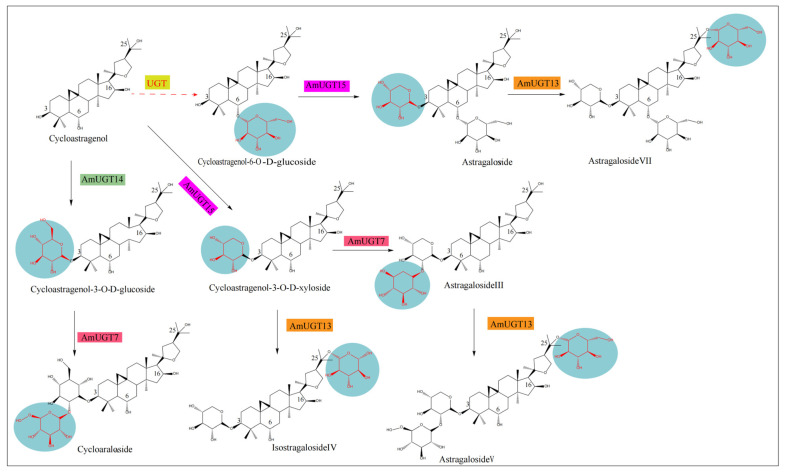
Astragaloside biosynthesis pathway. Note: The blue part represents different glycosyl groups, the pink, green and orange parts represent different glycosyltransferases. The red dotted line represents the speculative path.

**Table 1 plants-12-01858-t001:** *Astragalus* plant tissue culture research.

Culture Material	Culture Medium	Cultivation Conditions	Result
Callus	MS + 0.5 mg/L 6-BA + 2 mg/L2.4-D + 0.1 mg/LVC+3% sucrose + 0.7% agar	The culture temperature was 25 °C, 24 h dark culture, pH value was 5.8	The induction rate can reach 100% [16]
Suspension cell	MS + 0.1 mg/L (NAA) + 1.0 mg/L (6- BA) + 1.5% (*w/v*) sucrose and 0.8% (*w/v*) agar After three weeks, MS + 0.5 mg/L (IAA) + 1.5% (*w/v*) sucrose and 0.8% (*w/v*) agar	Seed germination, callus induction and subculture were carried out in a growth chamber illuminated with fluorescent light (ca. 1400 mol m^−2^ s^−1^) over a 16/8 day and night at 25 ± 2 °C	The seedlings developed fragile callus within 2 weeks [17]
Adventitious root	MS +7 mg/L(IBA)+ 30 g/L sucrose + 7 g/L agar	In dark, at 25 ± 2 °C, for 6 weeks of culture	Adventitious roots were successfully induced [18]

**Table 2 plants-12-01858-t002:** Study on metabolism control of *Astragalus*.

Culture Material	Activity Component	Influence Factor	Metabolic Regulation	Result
Suspension cells	PAL Activity and Total Phenol	Yeast extract (10 g/L) was added for 36 h on the 13th day of culture	PAL activity induced by yeast extract was positively correlated with total phenol accumulation	Increased PAL activity and total phenol content [30]
*Astragalus*	*Astragalus* saponin	---	HMGR, FPS, SE, and CAS are the main regulatory genes	Regulated the synthesis of *Astragalus* saponin [31]
Hairy roots	Saponins and Isoflavone	Regulation of Methyl Jasmonate (MJ), Acetylsalicylic Acid (ASA) and Salicylic Acid (SA)	MVD, IDI, FPS, SS, CHI, IFS	It is revealed that MVD, IDI, FPS and SS are key enzyme genes that MJ induces and which regulate the saponin biosynthesis pathway CHI and IFS are the key factors of the isoflavone biosynthesis pathway [21]
*Astragalus*	Cyclo*Astragalus* phenol and *Astragalus* phenol	Endophytic fungi	---	Endophytic fungi were found to transform sapogenins (Cycloenosterol and *Astragalus* cresol) [23]
Hairy roots of *Astragalus*	ASTS, MAO rhzomorphand CG	100 μM methyl jasmonate (MeJA) treatment	2127 genes were up-regulated by MeJA and 1247 genes were down-regulated by MeJA	The accumulation of ASTS, MAO rhizomorph and CG in hairy roots treated with MeJA increased significantly [32]
*Astragalus*	Genistein -7-O-β-D- glucoside (CGs)	Low temperature stress, light dependence	CHS, CHR, CHI, IFS, and I3’H PAL1, C4H	Temperature fluctuations up-regulated the transcription of CHS, CHR, CHI, IFS, and I3’H, but had different effects on the transcription of PAL1 and C4H of phenylpropanoid pathway in leaves [33]
Hairy roots	Hairy cephalosporins (CA) and formononetin (FO)	AMHRCs were co-cultured with immobilized aspergillus niger (IAN) for 54 h	---	The CA and FO biosynthetic pathway gene expression was significantly up-regulated, thereby increasing the production of CA and FO [34]
Root	Isoflavone	After 10 days of UV-B treatment (λ = 313 nm, 804 j/m)	---	UV-B radiation significantly induced isoflavone synthesis [35]
Hairy roots	Isoflavone	Ultraviolet light (UV-A, UV-B and UV-C) irradiation	PAL, C4H	86.4 kJ/m (2) UV-B upregulated the transcription and expression of all genes involved in the isoflavone biosynthesis pathway of AMHRCs [36]

**Table 3 plants-12-01858-t003:** Effect of Inducer on Secondary Metabolites of *Astragalus*.

Culture Material	Active Ingredient	Influence Factor	Increase Multiples
*Astragalus* adventitious roots	Calycosin isoflavone glycoside	Hydrogen peroxide, the L-phenylalanine	The culture treated with hydrogen peroxide and L-phenylalanine was 8.6 times higher than that treated with hydrogen peroxide alone [37]
*Astragalus* adventitious roots	Calycosin isoflavone glycoside	Drought stress, methyl jasmonate, and L-phenylalanine	The three combinations induced the highest CG content, 3.12 times higher than that of the field plants [18]
*Astragalus* hair root	*Astragalus* saponin I, *Astragalus* saponin and *Astragalus* methyside	Methyl jasmonate	It reached 2.98, 2.85, 2.30, and 1.57 times in the control group, respectively [32]
*Astragalus* hair root	*Astragalus* methylside	Chitosan	It was 2.1 times higher than that in the control group [38]

## Data Availability

No new data were created or analyzed in this study. Data sharing is not applicable to this article.

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
