# Peer review of "Advances in Biotechnological Production and Metabolic Regulation of Astragalus membranaceus"

_plants, 2023, doi:10.3390/plants12091858_

Round 1

Reviewer 1 Report

The manuscript is well structured and written at a high scientific level. I congratulate my colleagues for the hard work and, above all, for the result. I will only allow myself to recommend that the authors add, as supplementory  file/files, some pictures of the intact plant, its in vitro cultures and its bioreactor cultivation.

Author Response

Response to the Reviewers

Thank you very much for your kind review and give us good revision comments. We have carefully revised the manuscript according to the reviewers’ suggestions. The revised portions have been labeled in the resubmission manuscript.

Reviewer(s)' Comments to Author:

Reviewer: 1

Comments to the Author

Question: The manuscript is well structured and written at a high scientific level. I congratulate my colleagues for the hard work and, above all, for the result. I will only allow myself to recommend that the authors add, as supplementory  file/files, some pictures of the intact plant, its in vitro cultures and its bioreactor cultivation.

Answer:Thank you very much for your affirmation, which gives us great encouragement. The following are some supplementary pictures. If there are any problems, please make suggestions.

please see the word

Reviewer 2 Report

This review requires minor corrections:

-          first of all Latin names of organisms etc. should be written in italics – line 41, 49, 50, 72, 77, 81, 92, 100, 112, 119 (in vitro), 137, 143, 146, 185, 201, 220, 240, 247, 274, 275, 334, 340, 354, 356, 358, 362, 371, 382, 384, 386, 392, and throughout all the references;

-          line 44 instead of “isoflavone saponins” should be “isoastragalosides”;

-          line 96, the sentence “Optimal liquid culture conditions.” is unnecessary;

-          in some cases, a space is missing and a colon should be used instead of a comma or semicolon: line 72, 73, 103, 114, Table 2, Table 3, 230, 232, Note under Figure 1, 278, 283, 287, 305, - 311, 322, 330, 333, 370, 383;

-          line 119: should be “plants and plant cells” instead of “plant and plant cells”;

-          Table 2, the word formononetin should be lowercase and in brackets (FO) uppercase;

-          Table 3, the phrase  Astragalus saponins” is repeated and should be “Astragalus methylside”;

-          line 163-165 and 175-77, this sentence is unclear, the verb is missing;

-          line 184, unnecessary word “membranous”;

-          line 199, what is about the word “Body”?

-          line 258, there is an error in the notation of NAA (no space);

-          Figure 2; should be “cycloartenol” instead of “cyscloartenol”;

-          references, line 465, there is a mistake in the title.

Author Response

Response to the Reviewers

Thank you very much for your kind review and give us good revision comments. We have carefully revised the manuscript according to the reviewers’ suggestions. The revised portions have been labeled in the resubmission manuscript.

Reviewer(s)' Comments to Author:

Reviewer: 2

Comments to the Author

  first of all Latin names of organisms etc. should be written in italics – line 41, 49, 50, 72, 77, 81, 92, 100, 112, 119 (in vitro), 137, 143, 146, 185, 201, 220, 240, 247, 274, 275, 334, 340, 354, 356, 358, 362, 371, 382, 384, 386, 392, and throughout all the references;

-  line 44 instead of “isoflavone saponins” should be “isoastragalosides”;

-  line 96, the sentence “Optimal liquid culture conditions.” is unnecessary;

-  in some cases, a space is missing and a colon should be used instead of a comma or semicolon: line 72, 73, 103, 114, Table 2, Table 3, 230, 232, Note under Figure 1, 278, 283, 287, 305, - 311, 322, 330, 333, 370, 383;

-  line 119: should be “plants and plant cells” instead of “plant and plant cells”;

- Table 2, the word formononetin should be lowercase and in brackets (FO) uppercase;

- Table 3, the phrase  “Astragalus saponins” is repeated and should be “Astragalus methylside”;

- line 163-165 and 175-77, this sentence is unclear, the verb is missing;

- line 184, unnecessary word “membranous”;

- line 199, what is about the word “Body”?

- line 258, there is an error in the notation of NAA (no space);

- Figure 2; should be “cycloartenol” instead of “cyscloartenol”;

- references, line 465, there is a mistake in the title.

Answer:Thank you for your careful review. All the mistakes have been corrected in the paper. Please check the revised draft.

If there are still inappropriate places, welcome to point out.we would like very much to modify them and we really appreciate your help.

Reviewer 3 Report

Ji et al review the most relevant experimental data on the biotechnologies research of A. membranaceus. This review article is important and provides a valuable resource to the community. However, this paper is more like a collection of summaries of other papers, rather than a review paper. Authors should take a more critical view on the data and briefly outline the lines of future research that is needed.

1. line 77-80. Some discussion is required to explain the differences in the regeneration rate among organs.

2. line 92-99. Is there any difference in hairy root induction rate depending on the Agrobacterium rhizogenes strain?

3. What are the differences in regeneration, suspension, and adventitious root conditions compared to other plants belonging to the genus Astragalus?

4. line 168-171. "To study the response of hairy root of Astragalus to 100 μM methyl jasmonate (MeJA), DESeq analysis showed that 2127 genes were up-regulated, 1247 genes were down-regulated, and 17 new astragaloside biosynthesis genes and 7 were isolated new genes for the biosynthesis of isoflavones and isoflavone glucoside-7-O-β-D-glucoside." So what? The authors should provide the reader with information about which genes were present in the 2127 genes and what to expect from increased expression of those genes.

5. There are many differences or inconsistencies between the citations (references) and the text. For example, "Yangyang Duan et al. [57] found for the first time that AmCAS1 (a cycloartenol synthase) can catalyze the conversion of 2,3-oxidosqualene to cycloartenol, by transcriptome and phylogenetic analysis (line 315-317)." Reference 57 is a paper by Li et al. (2014), and there is no description about AmCAS1.

6. First time you use an abbreviation in the text, please present both the spelled-out version and the short form. For example, KT (line 83)?

7. Please check carefully for Citation format of references in the text. For example, "Li Xiaohua et al. [42] found that the…" must be changed to " Li et al. (2022) found that the…"

Author Response

Response to the Reviewers

Thank you very much for your kind review and give us good revision comments. We have carefully revised the manuscript according to the reviewers’ suggestions. The revised portions have been labeled in the resubmission manuscript.

Reviewer(s)' Comments to Author:

Reviewer: 3

Comments to the Author

  1. line 77-80. Some discussion is required to explain the differences in the regeneration rate among organs.

Thank you for your suggestion,We have reviewed relevant literature, and some of them did not mention induction rate, but only described the success of induction. The references to induction rate in literature have been shown in the paper.

2.line 92-99. Is there any difference in hairy root induction rate depending on the Agrobacterium rhizogenes strain?

Thank you for your question, We explain as follows: In the literature, hairy roots of astragalus were induced by Agrobacterium infection only, that is Agrobacterium rhizogenes LBA9402,Then, the hairy root strain AMHRLVI with high saponins and the hairy root strain AMHRLII with high isoflavones were screened based on the hairy root growth amount and the accumulation amount of secondary metabolic active components, instead of using two different agrobacterium tumefaciens.

3.What are the differences in regeneration, suspension, and adventitious root conditions compared to other plants belonging to the genus Astragalus?

Thank you for your question. There are about 449 plants in the genus Astragalus, but most of the studies on explants induced by  genus Astragalus only include Astragalus membranaceus (Fisch.) Bge. var. mongholicus (Bge.)Hsiao or Astragalus mem-branaceus (Fisch.) Bge,which cannot form a relatively accurate rule, However, for most plants, IBA is the most suitable hormone for inducing adventitious roots, and callus should be induced first to induce suspension cells. 2,4-D and NAA are the most suitable hormones for callus induction. Different plant explants have different sensitivity to different hormones. It is easier to induce success by selecting hypocotyls or leaves as explants, but the concentration and ratio of hormones have not been determined, which still needs to be proved by experiments.

  1. line 168-171. "To study the response of hairy root of Astragalus to 100 μM methyl jasmonate (MeJA), DESeq analysis showed that 2127 genes were up-regulated, 1247 genes were down-regulated, and 17 new astragaloside biosynthesis genes and 7 were isolated new genes for the biosynthesis of isoflavones and isoflavone glucoside-7-O-β-D-glucoside." So what? The authors should provide the reader with information about which genes were present in the 2127 genes and what to expect from increased expression of those genes.

Thank you for your question, We explain as follows:“17 new astragaloside biosynthesis genes and 7 were isolated new genes for the biosynthesis of isoflavones and isoflavone glucoside-7-O-β-D-glucoside.” The most available key genes in the literature belong to 2,127 up-regulated genes,This is what we did not describe clearly, we have added in the paper.

5.There are many differences or inconsistencies between the citations (references) and the text. For example, "Yangyang Duan et al. [57] found for the first time that AmCAS1 (a cycloartenol synthase) can catalyze the conversion of 2,3-oxidosqualene to cycloartenol, by transcriptome and phylogenetic analysis (line 315-317)." Reference 57 is a paper by Li et al. (2014), and there is no description about AmCAS1.

Thank you for your careful review. We have corrected it

6.First time you use an abbreviation in the text, please present both the spelled-out version and the short form. For example, KT (line 83)?

Thank you for your careful review, we have added in the article.

  1. Please check carefully for Citation format of references in the text. For example, "Li Xiaohua et al. [42] found that the…" must be changed to " Li et al. (2022) found that the…"

Thank you for your careful review, we have added in the article.

If there are still inappropriate places, welcome to point out.we would like very much to modify them and we really appreciate your help.

Round 2

Reviewer 3 Report

Accept in present form